# Clinical and economic outcomes of adding [18F]FES PET/CT in estrogen receptor status identification in metastatic and recurrent breast cancer in the US

Regina Munter-Young[1]*, Adolfo Fuentes-Alburo[2], Nicholas DiGregorio[3], Kurt Neeser[4], Dmitry Gultyaev[4]

1 Global Market Access, GE HealthCare, Marlborough, MA, United States of America, 2 Global Medical Affairs, GE HealthCare, Marlborough, MA, United States of America, 3 Medical Affairs, GE HealthCare, Marlborough, MA, United States of America, 4 Certara Germany GmbH, Evidence and Access, Loerrach, Germany

* regina.young@gehealthcare.com

## Abstract

### Background and objectives

Correct identification of estrogen receptor (ER) status in breast cancer (BC) is crucial to optimize treatment; however, standard of care, involving biopsy and immunohistochemistry (IHC), and other diagnostic tools such as 2-deoxy-2-[18F]fluoro-D-glucose or 2-[18F]fluoro-2-deoxy-D-glucose ([18F]FDG), can yield inconclusive results. 16α-[18F]fluoro-17β-fluoroestradiol ([18F]FES) can be a powerful tool, providing high diagnostic accuracy of ER-positive disease. The aim of this study was to estimate the budget impact and cost-effectiveness of adding [18F]FES PET/CT to biopsy/IHC in the determination of ER-positive status in metastatic (mBC) and recurrent breast cancer (rBC) in the United States (US).

### Methods

An Excel-based decision tree, combined with a Markov model, was developed to estimate the economic consequences of adding [18F]FES PET/CT to biopsy/IHC for determining ER-positive status in mBC and rBC over 5 years. Scenario A, where the determination of ER-positive status is carried out solely through biopsy/IHC, was compared to scenario B, where [18F]FES PET/CT is used in addition to biopsy/IHC.

### Results

The proportion of true positive and true negative test results increased by 0.2 to 8.0 percent points in scenario B compared to scenario A, while re-biopsies were reduced by 94% to 100%. Scenario B resulted in cost savings up to 142 million dollars.

### Conclusions

Adding [18F]FES PET/CT to biopsy/IHC may increase the diagnostic accuracy of the ER status, especially when a tumor sample cannot be obtained, or the risk of a biopsy-related

**Data Availability Statement:** All relevant data are within the manuscript and its Supporting Information files.

**Funding:** This study was funded by GE HealthCare. The funder provided support in the form of salaries for authors RM-Y, AF-A, and ND and funded the health economic analysis prepared by Certara Germany GmbH but did not have any additional role in the study design, data collection and analysis, decision to publish, or preparation of the manuscript. The specific roles of these authors are articulated in the 'author contributions' section.

**Competing interests:** RM-Y, AF-A, and ND are employees of GE HealthCare. KN and DG are employees of Certara Germany GmbH while Shereen Cynthia D'Cruz, Ph.D. an employee of Certara Synchrogenix. Certara is a consulting company that was paid by GE HealthCare for the services rendered. This does not alter our adherence to PLOS ONE policies on sharing data and materials.

complication is high. Therefore, adding [18F]FES PET/CT to biopsy/IHC would have a positive impact on US clinical and economic outcomes.

## Introduction

Breast cancer (BC) is the most diagnosed cancer worldwide, with approximately 2.3 million new cases and 685 000 deaths in 2020 [1]. Around 70% of BCs express estrogen receptors (ER) [2–4], which is one of the criteria that helps characterize the disease and identify an optimal treatment pathway [5, 6]. Biopsy and immunohistochemistry (IHC) are the standard of care [5, 7] to determine the ER status of any BC, including metastatic (mBC, ie, first occurrence of metastases) and recurrent BC (rBC, ie, reoccurrence of the disease after a period of time when it was under control). However, given the following limitations of biopsy and IHC, there remain several unmet needs.

The first unmet need impacts the confidence one might have in the diagnostic accuracy of a single-location biopsy/IHC [8, 9], given differences within one tumor and across multiple tumors' characteristics, known as intra-tumor and inter-tumor heterogeneity, respectively. Intra-tumoral heterogeneity can be further differentiated into spatial and temporal heterogeneity. In spatial heterogeneity, different areas within the tumor may present different tumor characteristics (eg, phenotypic, epigenetic) simultaneously. Temporal heterogeneity, on the other hand, is when tumor characteristics change because of tumor progression over time. In addition, heterogeneity presents between the primary tumor and metastases, as well as among different metastatic lesions, which is known as intra-metastatic heterogeneity. The cause of heterogeneity within the tumor or metastases can be attributed not only to tumor progression but also to therapy-related clonal selection. This effect can be attributed to the fact that primary cancer types, unlike metastases, exhibit higher clonal variability (mutations and structural changes) due to the selection of oncological therapies [10]. The discordances in ER status within and/or across different lesions can result in false positive or negative biopsy/IHC results [9]. A second unmet need limiting the confidence in the diagnostic accuracy of a single-location biopsy/IHC [8, 9] is the fact that some metastatic lesions are difficult to access, such as those in the brain [3]. The common practice is to target the lesion to which there is the easiest access and low risk of complications, rather than the lesion with the most clinical significance [11, 12]. A third limitation of biopsy/IHC which leads to inconclusive results is the inability to retrieve an adequate specimen which may be the result of bleeding, bone decalcification, or some other technical reasons, eg, due to sampling an inappropriate tissue [9, 13, 14]. The three highlighted limitations are associated with a risk of analyzing non-representative tissue, not characterizing the full extent of the ER status, which may result in misdiagnosis and the administration of inappropriate therapies with sub-optimal clinical response [15, 16]. A fourth limitation of biopsy/IHC is that despite a positive test result for biopsy/IHC, up to 50% of ER-positive BCs as identified through biopsy/IHC will not respond to endocrine therapy. This may be because not all ERs detected by IHC are available or functionally available for binding to their estrogen ligands [17]. While positron emission-tomography (PET) using 2-deoxy-2-[18F]fluoro-D-glucose or 2-[18F]fluoro-2-deoxy-D-glucose ([18F]FDG) is also a widely accepted imaging technique for staging mBC and assessing the effectiveness of treatment [18], it is also associated with several limitations. [18F]FDG PET uptake reflects the metabolic activity of tissues, posing a challenge in its ability to detect metastases with low metabolic rates, such as tumors associated with invasive lobular carcinoma (ILC) [19]. In addition, [18F]FDG PET has limited sensitivity for brain metastases due to high background activity, which could

reduce signal to-noise ratio [20]. Another limitation of [18F]FDG PET is that it is not specific to a single biomarker and detects infections and inflammation, both of which may result in inappropriate interpretations of the diagnostic procedure [21]. Finally, [18F]FDG PET, Computed Tomography (CT), and Bone Scintigraphy might yield inconclusive or non-diagnostic results, as findings could be attributed to either malignancy or benign degenerative changes.

16α-[18F]fluoro-17β-fluoroestradiol ([18F]FES) is a radiolabeled form of estrogen that binds to both alpha and beta ER, for use with PET imaging/CT. This compound allows for a minimally invasive yet selective [22] and comprehensive assessment of the whole-body ER-positive tissue status. In the case of multi-metastatic disease, when the standard of care would require only one metastasis to be biopsied, [18F]FES can help to evaluate the ER status of the non-biopsied metastases [17].

As determined in prospective trials and meta-analyses, [18F]FES PET/CT provides high diagnostic accuracy of ER-positive disease, which can be used to supplement the lesion level knowledge offered by IHC from tissue sampling. Furthermore, a review of 6 clinical trials showed that [18F]FES PET/CT identified a greater number of metastatic lesions compared to [18F]FDG PET [23]. This may inform potential therapy adjustments leading to improvement of the quality and effectiveness of the treatment in patients with mBC and rBC [3, 9]. Beyond that, [18F]FES PET/CT may provide higher diagnostic accuracy of ER-positive disease for clinical dilemmas such as an inability to determine extent of (suspected) metastatic disease with standard workup, unclear ER status of the tumor, and an inability to determine which primary tumor caused metastases [24].

The aim of this study was to estimate the budget impact and cost-effectiveness of the addition of [18F]FES PET/CT to biopsy/IHC in the determination of ER-positive status in mBC and rBC in the United States (US).

## Methods

### Model description and settings

An Excel-based decision tree combined with a Markov model was developed to estimate the economic consequences over a 5-year period in the US. To perform the economic analyses, the following scenarios were compared: Scenario A, where [18F]FES PET/CT is not available and the determination of ER-positive status in patients with mBC and rBC is carried out solely through biopsy and IHC; Scenario B, where [18F]FES PET/CT is used in addition to biopsy/IHC.

### Target population estimation

Epidemiological data were obtained from the published literature and presented in Table 1. Initially, this economic analysis aimed to include all US patients with mBC and rBC; however, the underlying clinical studies examining FES excluded 2 types of patients: those who had liver metastases, as [18F]FES PET/CT is taken up in the liver, and those receiving palliative care [2, 24]. The models assumed that [18F]FES PET/CT was added to biopsy/IHC to accurately classify the ER status in 3 patient groups: (i) mBC patients when biopsy failed or was inconclusive, (ii) mBC patients when biopsy was not possible, (iii) all rBC patients [3, 25, 26]. Different subgroups according to location of the metastases (ie, visceral, skeletal, cerebral, and spinal column) and human epidermal growth factor receptor 2 (HER2) status (HER2-negative, HER2-positive, and any HER2) were also analyzed.

### Clinical inputs

[18F]FES was produced locally as described by Roemer et al. and Venema et al. [49, 50]. The [18F]FES PET/CT scans were performed on a BioGraph mCT 40- or 64-slice PET/CT

**Table 1. Inputs.**

| Epidemiological data and clinical input assumptions | |
|---|---|
| **Demographics** | |
| Average age (SD) in years | 58 (+/-13) [27] |
| Age min, max in years | 18, 100 [27] |
| **Proportion of patients regarding HER2 status** | |
| HER2-negative | 88% [28] |
| HER2-positive | 12% [28] |
| **Number of lesions per patient (HER2-negative, HER2-positive)** | |
| Single lesion | 43%, 22% [29] |
| 2 lesions | 29%, 30% [29] |
| 3 lesions | 18%, 30% [29] |
| 4 lesions | 7%, 15% [29] |
| 5+ lesions | 3%, 3% [29] |
| **Metastatic location distribution (HER2-negative, HER2-positive)** | |
| Liver | 17%, 31% [28] |
| Lungs | 23%, 50% [28] |
| Bones | 56%, 63% [28] |
| Brain | 6%, 6% [28] |
| Other | 57% [30], 57% [31] |
| **Discordance rates** | |
| HER2-negative | 27% [32] |
| HER2-positive | 30% [32] |
| **ER status** (n = 180): +/- rate (1% ER expression threshold) | |
| IHC of biopsied lesion | 134, 46 [26] |
| Whole body [18F]FES PET result | 126, 38 [26] |
| **Sensitivity (SE)** | |
| [18F]FES PET/CT | 0.86 (+/-0.09) [33] |
| IHC | 0.99 (+/-0.10) [34] |
| **Specificity (SE)** | |
| [18F]FES PET/CT | 0.85 (+/-0.08) [33] |
| IHC | 0.93 (+/-0.09) [34] |
| **Biopsy/IHC failure rate** | |
| Biopsy—All lesions | 20% [35] |
| Biopsy—Single lesion | 20% [35] |
| Biopsy—Visceral metastases | 20% [35] |
| Biopsy—Skeletal metastases | 23% [36] |
| Biopsy—Brain | 23% * |
| Biopsy—Spine | 23% [36] |
| IHC failure rate | 2% [37] |
| **Probability of progression per cycle (1st line, 2nd line, 3rd+ line)** | |
| Inappropriate chemotherapy | 12%, 17%, 24% & |
| Appropriate endocrine therapy | 5%, 7%, 10% & |
| Appropriate chemotherapy | 9%, 12%, 19% & |
| Inappropriate endocrine therapy | 14%, 20%, 27% & |
| **Rate of treatment-related AEs (AEs on chemotherapy, AEs on endocrine therapy)** | |
| Anemia | 10.0% [38], 0.7% [39] |
| Diarrhea | 47.0% [38], 0.0%* |
| Fatigue | 12.0% [38], 0.0%* |

(*Continued*)

**Table 1.** (Continued)

| Epidemiological data and clinical input assumptions | |
|---|---|
| Hepatobiliary toxicity | 1.0% [40], 5.8% [41] |
| Severe neutropenia | 0.0% [38], 0.9% [39] |
| Vomiting | 15.0% [38], 0.0%* |
| **Cost inputs and utilities (CEM only)** | |
| **Description** | |
| **Costs (USD)** | |
| [18F]FES PET/CT | 4650* |
| Biopsy–liver | 1196 [42] |
| Biopsy–lung | 4356 [43] |
| Biopsy–bone | 2180 [44] |
| Biopsy–brain | 10 000 * |
| Biopsy–other | 1228 [45] |
| Chemotherapy per cycle (1st, 2nd, 3rd+ line) | 1199, 2398, 2012 [46] |
| Endocrine therapy per cycle (1st, 2nd, 3rd+ line) | 12 886, 10 961, 10 961 [46] |
| Anemia | 6779 [40] |
| Diarrhea | 4809 [40] |
| Fatigue | 0.00 [47] |
| Hepatobiliary toxicity | 5915 [41] |
| Severe neutropenia | 17 181 [41] |
| Vomiting | 4809 [40] |
| **Utilities per health state (CEM only)** | |
| On-treatment | 0.715 [48] |
| Progression (1st line) | 0.443 [48] |
| Progression (2nd line) | 0.230 [48] |
| Palliative treatment | 0.230 * |
| Death | * |

[18F]FES PET/CT indicates 16α-18F-fluoro-17β-fluoroestradiol with positron emission tomography/computed tomography; AEs, adverse events; CEM, cost-effectiveness model; IHC, immunohistochemistry; HER2, human epidermal growth factor receptor 2; SD, standard deviation; SE, standard error. *Assumption; & calculated

(Siemens/CTI, Knoxville, TN) or an Ingenuity TF or Gemini TF PET/CT (Philips Medical Systems, Cleveland, OH) where about 200 MBq [18F]FES were injected intravenously as a bolus. As described by van Geel et al., PET/CT images were acquired 60 ± 10 minutes after tracer injection [26]. Before the PET scan, a low-dose CT was conducted for attenuation and scatter correction. A whole-body PET emission scan was obtained with a 3-minute per bed position acquisition time. All patients were scanned from head to mid-thigh and for all scan reconstructions and quantifications, the European Association of Nuclear Medicine Research Limited (EARL) criteria were followed [51]. A full account of [18F]FES PET/CT whole body evaluation and the quantity of FES taken up by the metastases is further described by van Geel et al.

To assess the economic impact associated with the determination of ER-positive status in patients with mBC and rBC, the ER discordance rates, diagnostic sensitivity and specificity, biopsy- and IHC failure rates, and the frequency of adverse events (AEs) were considered (Table 1). ER discordance rates were used to calculate the proportion of patients with mBC who switched from ER-positive to ER-negative during the metastatic phase. The discordance rates in this model were calculated using a published retrospective analysis based on data

collected from patients who underwent surgery for BC between 2001 and 2014. In this study, the ER status of 132 patients was determined by pathologists as per ASCO (American Society of Clinical Oncologists) guidelines and the discordances were observed [32]. The sensitivity and specificity of [18F]FES PET/CT and IHC were used as the primary measures of diagnostic efficacy to calculate the true positive and true negative results of ER status. The inputs for [18F] FES PET/CT were informed by a meta-analysis [33] and those of IHC were sourced from the published literature [34]. A meta-analysis by Mo et al. in 2021 assessed the sensitivity and specificity of [18F]FES PET/CT in detecting ER expression in patients with recurrent or metastatic breast cancer [33]. The researchers conducted a comprehensive search of databases for relevant studies published between January 1980 and May 15, 2020. This search yielded eight studies that evaluated diagnostic tests.

In the selection and categorization of these studies, the researchers collaborated with other investigators to resolve any issues or disagreements. They organized the studies by their characteristics, types, and the reliability of their methods. The data from these studies were then compiled into a 2x2 table.

Using MetaDisc 1.4, a statistical software, the researchers performed a meta-analysis. They used the I2 statistic to evaluate heterogeneity. These analyses considered the type of intervention tests, such as PET/CT, and factors like metastasis, recurrence, and metastasis sites.

The meta-analysis aimed at evaluating diagnostic accuracy of [18F]FES PET/CT showed that the overall sensitivity and specificity were 0.86 and 0.85, respectively. This was further confirmed by van Geel et al. showing sensitivity and specificity 0.86 and 0.86 respectively when comparing [18F]FES PET and ER status by IHC [26].

Based on the results of the diagnostic testing, the model allowed for patients to be given either endocrine or chemotherapy treatments. Treatments were identified as "appropriate", ie, expected to produce a therapeutic effect or "inappropriate", ie, not expected to produce a therapeutic effect according to the clinical practice guidelines [52, 53]. Biopsy- and IHC failure rates, which were used as a proxy for the probability that the ER status cannot be determined from a biopsy [35, 36], and the frequency of AEs were sourced from the published literature [54].

## Costs

To establish the total cost per scenario, the model considered the cost of the molecular imaging tracer and any diagnostic imaging scans, the cost of biopsies, treatment costs, costs of managing treatment-related adverse events (TRAEs), and other relevant categories of cost, such as end-of-life care. All costs were inflated to 2021 US dollars (Table 1). The cost of [18F]FES was based on the wholesale acquisition cost (WAC), while the costs of other diagnostic imaging scans, including PET/CT, bone CT, and [18F]FDG scan, as well as magnetic resonance imaging (MRI), were sourced from the published literature [55]. The costs of biopsies were sourced from the Centers for Medicare & Medicaid Services (Hospital outpatient departments) and were differentiated by the organ examined (ie, liver, lungs, bones, brain, and others) [42–44]. The treatment costs included chemo- and endocrine therapy in the 1st, 2nd, and 3rd lines [56], as well as costs associated with TRAEs, which were based on published sources [40, 41, 47, 54, 57]. These cost inputs were then multiplied by the number of patients in each subpopulation within each intervention to calculate the annual total costs of scenarios A and B.

## Results

### Proportion of true positive and true negative test results

The analysis suggests that the introduction of [18F]FES PET/CT (scenario B) may lead to an increase in the proportion of true positive and true negative test results (Table 2). Table 2

**Table 2. BIM results per patient group in considered scenarios.**

| | Year 1 | | Year 2 | | Year 3 | | Year 4 | | Year 5 | |
|---|---|---|---|---|---|---|---|---|---|---|
| Scenario | A | B | A | B | A | B | A | B | A | B |
| **Application (i) When biopsy failed or was inconclusive in mBC** | | | | | | | | | | |
| Total costs (in millions USD) | 755.2 | 760.1 | 1862.6 | 1856.6 | 2613.1 | 2603.4 | 3106.2 | 3094.4 | 3431.0 | 3417.8 |
| Incremental (B vs A) | 4.9 | | -6.0 | | -9.7 | | -11.8 | | -13.2 | |
| Proportion of true positives and true negatives | 86.3% | 86.6% | 88.1% | 88.3% | 89.0% | 89.2% | 89.5% | 89.7% | 89.8% | 89.9% |
| Incremental (B vs A, pp) | 0.3 | | 0.2 | | 0.2 | | 0.1 | | 0.1 | |
| Re-biopsies | 1512 | 59 | 1563 | 79 | 1604 | 89 | 1646 | 95 | 1656 | 99 |
| Incremental (B vs A, %) | -1453 (-96.1%) | | -1484 (-95.0%) | | -1515 (-94.5%) | | -1551 (-94.2%) | | -1556 (-94.0%) | |
| **Application (ii) When biopsy was not possible in mBC** | | | | | | | | | | |
| Total costs (in millions USD) | 755.2 | 711.0 | 1862.6 | 1760.0 | 2613.1 | 2492.6 | 3106.2 | 2977.1 | 3431.0 | 3295.5 |
| Incremental (B vs A) | -44.2 | | -102.6 | | -120.5 | | -129.1 | | -135.5 | |
| Proportion of true positives and true negatives | 86.3% | 88.2% | 88.1% | 89.5% | 89.0% | 90.2% | 89.5% | 90.6% | 89.8% | 90.8% |
| Incremental (B vs A, pp) | 1.9 | | 1.4 | | 1.2 | | 1.1 | | 1.0 | |
| Re-biopsies | 1512 | 1514 | 1563 | 1565 | 1604 | 1606 | 1646 | 1648 | 1656 | 1658 |
| Incremental (B vs A, %) | 1 (0.1%) | | 2 (0.1%) | | 2 (0.1%) | | 2 (0.1%) | | 2 (0.1%) | |
| **Application (iii) When [18F]FES PET/CT in combination with IHC for rBC patients** | | | | | | | | | | |
| Total costs (in millions USD) | 229.9 | 208.3 | 460.3 | 418.6 | 562.9 | 519.2 | 617.9 | 571.6 | 659.7 | 610.5 |
| Incremental (B vs A) | -21.6 | | -41.7 | | -43.8 | | -46.4 | | -49.2 | |
| Proportion of true positives and true negatives | 86.5% | 91.9% | 88.1% | 92.8% | 88.6% | 93.0% | 88.7% | 93.1% | 88.9% | 93.2% |
| Incremental (B vs A, pp) | 5.4 | | 4.7 | | 4.4 | | 4.4 | | 4.3 | |
| Re-biopsies | 40 | 0 | 40 | 0 | 42 | 0 | 44 | 0 | 44 | 0 |
| Incremental (B vs A, %) | -40 (-100%) | | -40 (-100%) | | -42 (-100%) | | -44 (-100%) | | -44 (-100%) | |

[18F]FES PET/CT indicates 16α-[18F]-fluoro-17β-fluoroestradiol with positron emission tomography imaging/computed tomography; BIM, budget impact model; IHC, immunohistochemistry; mBC, metastatic breast cancer; pp, percentage point; rBC, recurrent breast cancer.

quantifies the economic, diagnostic, and clinical benefits of adding [18F]FES PET/CT to the diagnostic workup in mBC patients when a biopsy failed, was inconclusive, or not possible. Table 2 also quantifies the extent to which [18F]FES PET/CT in combination with IHC benefits rBC patients. When biopsy failed or was inconclusive in mBC patients, the sum of estimated true positive and true negative test results in year 1 were 86.3% in scenario A, compared to 86.6% in scenario B. The increase in diagnostic accuracy from adding [18F]FES PET/CT to biopsy/IHC led to a higher proportion of true results over the 5-year time horizon where the proportion of true positives and true negative test results were 89.8% in scenario A versus 89.9% in scenario B. A similar trend was observed when biopsy was not possible in mBC patients. The percentage of true test results over the 5-year time horizon was between 86.3% and 89.8% in scenario A, compared to a range of 88.2% to 90.8% in scenario B. The largest increase in diagnostic accuracy takes place when [18F]FES PET/CT is used in combination with biopsy/IHC in rBC patients (Table 2). Analysis suggests that prior to the introduction of [18F]FES, the proportions of true positive and true negative test results were 86.5% and 88.9% in year 1 and year 5, respectively. These values are expected to rise to 91.9% and 93.2% in the corresponding years with the inclusion of [18F]FES in biopsy/IHC.

In all patient groups, the benefits of adding [18F]FES PET/CT to biopsy/IHC are expected to be greater when patients present with multiple lesions. When biopsy was not possible in mBC patients who presented with 1 lesion, the proportion of true positive and true negative test results increased by 1.8 percentage points in year 1, while the same outcome rose by 4.0 percentage points if patients had more than 5 lesions. The advantage of introducing [18F]FES

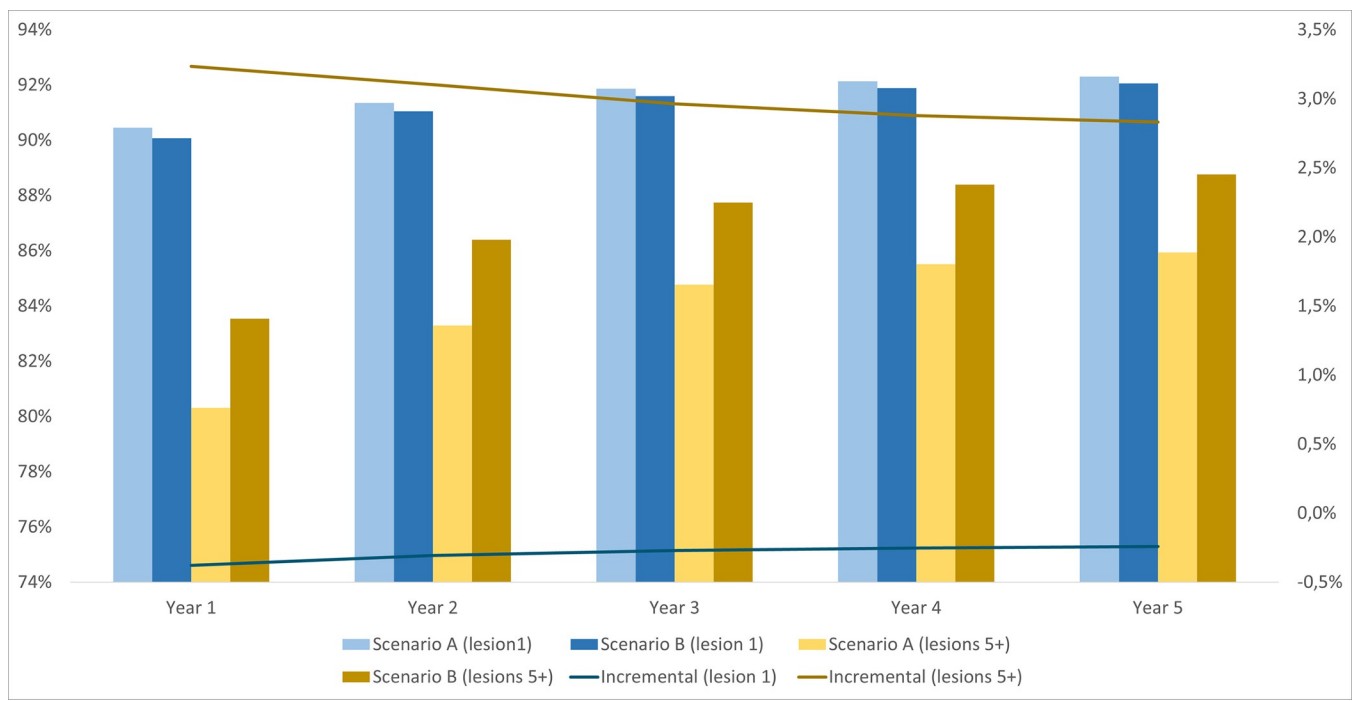

**Fig 1. The proportion of true positives and true negatives per number of lesions for mBC patients when biopsy failed or was inconclusive.**

PET/CT is particularly evident for rBC patients. The incremental diagnostic accuracy, measured by true results, ranged from 3.7 to 4.7 percentage points over the entire time horizon in patients with 1 lesion, while the corresponding outcome increase ranged from 5.1 to 6.3 percentage points in patients with 5 lesions (Figs 1–3).

## Number of re-biopsies

Introducing [18F]FES PET/CT decreases the number of re-biopsies performed except when biopsy was not possible (Table 2). In scenario A, when biopsy failed or was inclusive in mBC patients, the number of re-biopsies performed was estimated to be 1512 (year 1) and 1656 (year 5). The corresponding number of re-biopsies decreased to 59 (year 1) and 99 (year 5) in scenario B, thereby yielding a reduction of 1453 (-96.1%) and 1556 (-94.0%) in year 1 and year 5. The largest decrease in the number of re-biopsies was associated with rBC patients when [18F]FES PET/CT was used in combination with biopsy/IHC. In scenario A, the number of re-biopsies conducted varied between 40 and 44 during the entire period, whereas in scenario B, there were no expected re-biopsies over the entire time frame (Table 2).

## Total costs per scenario, budget impact, and cost-effectiveness

The most significant reduction in total costs was associated with mBC patients when biopsy was not possible. In scenario A, the total cost was estimated to be $755.2 million and $3431.0 million in year 1 and year 5, respectively. The corresponding expenditure decreased to $711.0 million and $3295.5 million, thereby yielding a savings of $44.2 million and $135.5 million in year 1 and year 5. Details of total costs for each scenario, stratified by patient group, can be found in Table 2 and S1–S3 Figs.

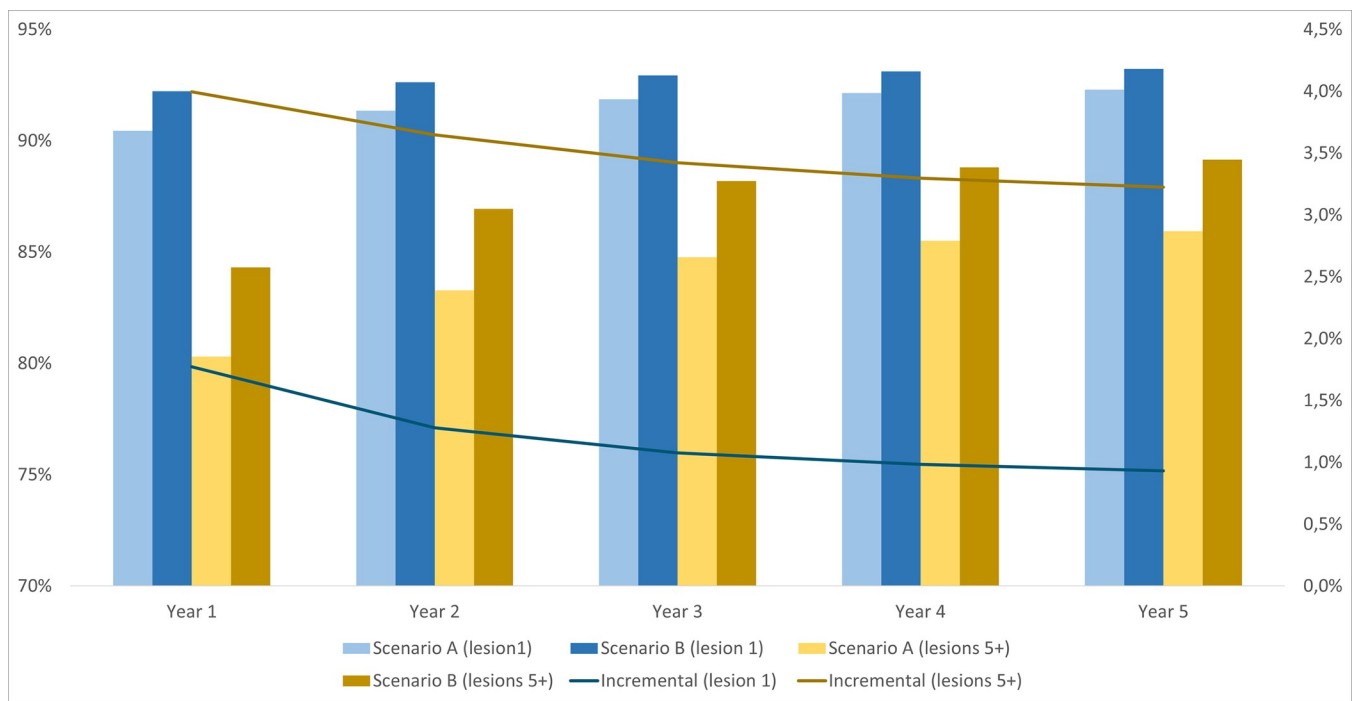

**Fig 2. The proportion of true positives and true negatives per number of lesions for mBC patients when biopsy was not possible.**

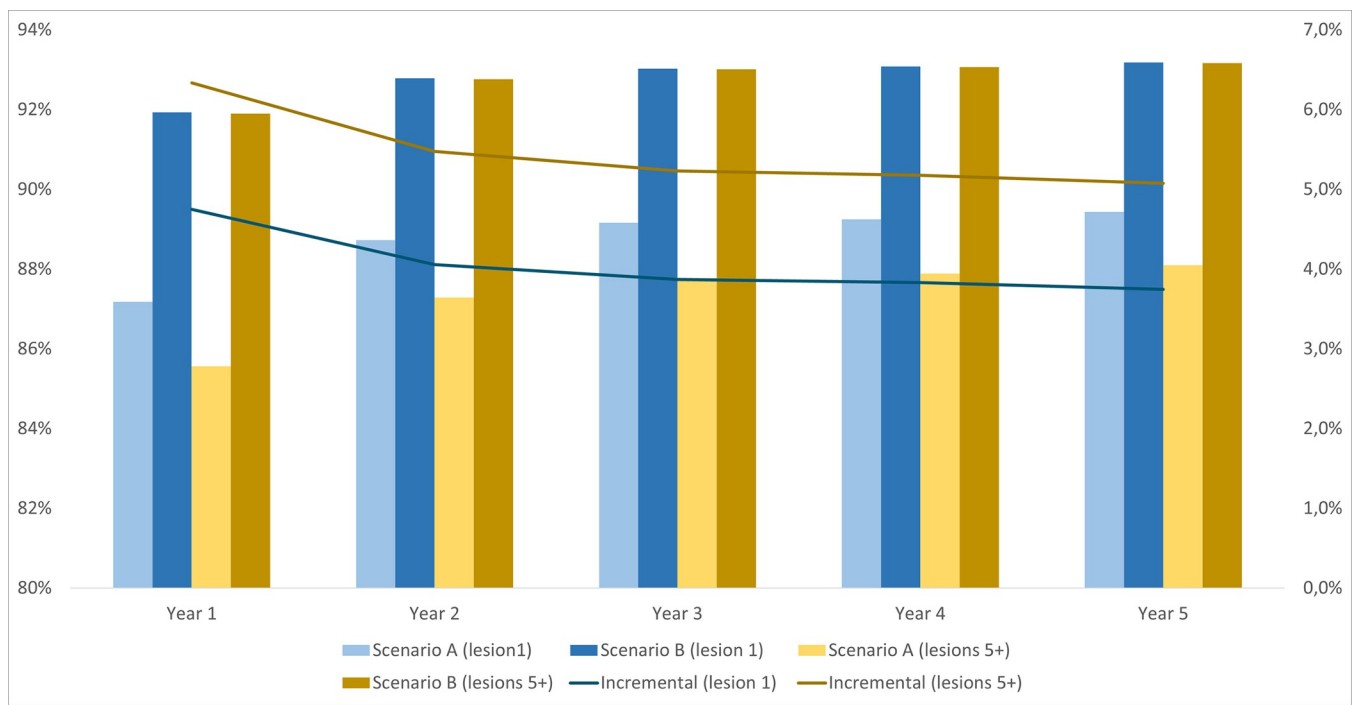

**Fig 3. The proportion of true positives and true negatives per number of lesions for rBC patients when [18F]FES PET/CT is used in combination with IHC.Abbreviations: [18F]FES PET/CT indicates 16α-18F-fluoro-17β-fluoroestradiol with positron emission tomography imaging/computed tomography; IHC, immunohistochemistry; mBC, metastatic breast cancer; rBC, recurrent breast cancer.**

**Table 3. CEM results over a five-year time horizon.**

| | [<sup></sup>$^{18}$F]FES PET/CT | Biopsy/IHC | Incremental Results |
|---|---|---|---|
| **Application (i) When biopsy failed or was inconclusive in mBC** | | | |
| Total costs (USD) | 282 734 | 283 790 | -1056 |
| Life years | 2.384 | 2.380 | 0.004 |
| QALYs | 1.281 | 1.278 | 0.004 |
| ICER | Biopsy/IHC is dominated by [$^{18}$F]FES PET/CT | | |
| **Application (ii) When biopsy was not possible in mBC** | | | |
| Total costs (USD) | 273 125 | 283 790 | -10 665 |
| Life years | 2.398 | 2.380 | 0.018 |
| QALYs | 1.281 | 1.278 | 0.003 |
| ICER | Biopsy/IHC is dominated by [$^{18}$F]FES PET/CT | | |
| **Application (iii) When [$^{18}$F]FES PET/CT in combination with IHC for rBC patients** | | | |
| Total costs (USD) | 113 530 | 120 969 | -7439 |
| Life years | 1.426 | 1.394 | 0.032 |
| QALYs | 0.448 | 0.444 | 0.004 |
| ICER | Biopsy/IHC is dominated by [$^{18}$F]FES PET/CT | | |

[$^{18}$F]FES PET/CT indicates 16α-$^{18}$F-fluoro-17β-fluoroestradiol with positron emission tomography imaging/ computed tomography; CEM, cost-effectiveness model; ICER, incremental cost-effectiveness ratio; mBC, metastatic breast cancer; QALYs, quality-adjusted life years; rBC, recurrent breast cancer.

Introducing [$^{18}$F]FES PET/CT to determine ER-positive status in mBC and rBC is cost effective as portrayed in Table 3. It was associated with a gain in life expectancy and quality-adjusted life years (QALYs) of 0.004 and 0.004 years when [$^{18}$F]FES PET/CT was applied after failed biopsy in mBC patients, 0.025 and 0.008 years when biopsy was not possible in mBC patients, and 0.018 and 0.003 years in rBC patients, respectively. In addition, [$^{18}$F]FES PET/CT was associated with a cost reduction compared with biopsy/IHC in each of the 3 scenarios (Table 3). Based on these results, it can be concluded that, in health economic terms, combining [$^{18}$F]FES PET/CT and biopsy/IHC dominates biopsy/IHC alone.

## Sensitivity analyses

Sensitivity analyses were carried out to investigate uncertainties surrounding model inputs as well as to test the robustness of the results. The one-way sensitivity analysis (OWSA) was performed by varying input parameters individually by a range of ±10%. The variables that affect the incremental cost-effectiveness ratio (ICER) most were the sensitivity of [$^{18}$F]FES PET/ CT in mBC patients when biopsy failed or was inconclusive and appropriateness of endocrine therapy in mBC patients when biopsy was not possible, and in rBC patients when [$^{18}$F]FES PET/CT was used in combination with IHC. The estimated ICER was also sensitive to changes in other clinical inputs, such as discordance rates, and specificity of [$^{18}$F]FES PET/CT and IHC. Other key drivers of changing the ICER were revealed and depicted in a tornado diagrams (S4–S6 Figs). S4–S6 Figs show the OWSA for the 10 most sensitive parameters in different subpopulations. S4 Fig identifies the parameters that impacts the ICER most among mBC patients when biopsy failed or was inconclusive. Similarly, S5 Fig illustrates the parameters that impact the ICER among mBC patients when biopsy was not possible, and S6 Fig shows them among rBC patients when [$^{18}$F]FES PET/CT is used in combination with IHC.

The probabilistic sensitivity analysis (PSA) was conducted with 1000 simulations to test the combined uncertainty and correlation between model parameters simultaneously. Mean

probabilistic results are presented in S1 Table. Biopsy/IHC are dominated by [$^{18}$F]FES PET/CT in all analyses, confirming the initial findings. The cost-effectiveness planes are illustrated in S7–S9 Figs to plot the incremental costs and effects of [$^{18}$F]FES PET/CT against biopsy/IHC. Most of the iterations fall in the southeast quadrant in all analyses, indicating that [$^{18}$F]FES PET/CT is more effective and less costly than biopsy/IHC.

## Discussion

To date, only a limited number of economic analyses have been performed for [$^{18}$F]FES PET/CT in this indication of mBC and rBC. Koleva-Kolarova et al. [58] evaluated the cost-effectiveness of [$^{18}$F]FES PET/CT and [$^{89}$Zr]Zr-Trastuzumab in comparison with biopsy/IHC for first-line treatment of mBC patients in the Netherlands. The ICER for replacing biopsy with [$^{18}$F]FES PET/CT imaging with FES and [$^{89}$Zr]Zr-Trastuzumab was cost-effective, ranging between €71,000 and €77,000 per life year gained. Sensitivity analyses have shown that even a small change in the sensitivity and specificity of [$^{18}$F]FES PET/CT has a significant impact on the cost-effectiveness ratio. Although this economic analysis was performed in a different country, it serves to reinforce the clinical and economic benefits of adding [$^{18}$F]FES PET/CT to the diagnostic workup in patients with mBC.

The OWSA has shown that the sensitivity of [$^{18}$F]FES PET/CT (S4) and the appropriateness of endocrine therapy (S5, S6) had the most significant impact on the ICER. This implies that these two factors have the highest impact on the cost-effectiveness of the healthcare intervention being analyzed. The sensitivity of a diagnostic test like [$^{18}$F]FES PET/CT refers to its ability to correctly identify ER-positive patients, while the appropriateness of endocrine therapy relates to the correct selection and application of hormone treatments for BC patients. High sensitivity in diagnostic imaging and accurately, tailored endocrine therapy can lead to better patient outcomes and more efficient use of healthcare resources. Other clinical inputs, such as the specificity of IHC or discordance rates, also influenced the ICER, but to a lesser extent as represented by the Tornado diagrams. The importance of this observation lies in its ability to provide guidance to healthcare decision makers. By identifying the factors that most influence the cost-effectiveness of [$^{18}$F]FES PET/CT in the treatment of mBC and rBC, resources can be allocated more efficiently, and attention can be focused on improving these key areas.

The visual representation of PSA using scatterplots shows that [$^{18}$F]FES PET/CT provides better health outcomes at lower costs for patients with mBC and rBC in most simulations compared to biopsy/IHC. The robustness of the results supports the argument for introducing [$^{18}$F]FES PET/CT into clinical practice for the subpopulations studied.

The aim of the study was to assess the budget impact and cost-effectiveness of the addition of [$^{18}$F]FES PET/CT to biopsy/IHC in the determination of ER-positive status in mBC and rBC in the US. The economic models developed in this study utilized data from various sources, including published meta-analyses. Key findings include that the introduction of [$^{18}$F]FES PET/CT in combination with biopsy/IHC is expected to improve the accuracy of diagnosis of patients' ER status. The increase in diagnostic accuracy is due to an increase in the proportion of true positive and true negative test results by 0.2 to 8.0 percent points compared to the scenario where it was not used. The analysis also showed that introducing [$^{18}$F]FES PET/CT may reduce re-biopsies by as much as 94% to 100% except in case where biopsy was not possible in mBC patients. Improved diagnostic accuracy may potentially result in cost savings up to 142 million dollars. This study therefore suggests that adding [$^{18}$F]FES PET/CT to biopsy/IHC could have a positive impact on clinical and economic outcomes as well as on patients´ quality of life (QoL) in the US.

From a clinical perspective, an increase in diagnostic accuracy represents an opportunity to produce a better therapeutic decision across a variety of metastatic conditions. In cases of multi-metastatic disease, typically only 1 or few lesions are biopsied increasing the possibility of heterogeneity both over time and across the number of lesions. However, a whole-body evaluation of the receptor status across all metastases would provide an accurate mapping of a patient's potential response to treatment. In oligometastatic disease (<5 lesions), while it might be possible in some cases to biopsy all of the metastases, there are instances where some lesions are difficult to access. [18F]FES PET/CT offers the opportunity to evaluate all lesions' ER status. In mono-metastatic disease, after confirming the ER status via biopsy/IHC, the use of [18F]FES PET/CT might appear superfluous. However, considering the number of false negative results in conventional studies such as mammography, MRI, or ultrasound for axilla in low-metabolic tumors (like ILC), the use of [18F]FES PET/CT may be justified [59–62]. Finally, in recurrent disease while the use of biopsy/IHC to confirm the ER status is not typically problematic, there is a growing body of evidence that questions the ability of conventional diagnostics and imaging to detect distant disease in low-metabolic diseases. Appropriate ER status characterization is essential to informing the most appropriate therapeutic plan and prognosis.

In addition to more reliable decision-making through [18F]FES PET/CT, patients may benefit directly from reduced disease progression, increased lifespan, and improved QoL. The studies by Boers et al. and Liu et al. have shown that ER-positive/HER2-negative mBC patients treated with Palbociclib, a CDK4/6 inhibitor plus endocrine therapy have a higher progression-free survival (PFS) than patients with ER-negative or uncertain ER status. As a result, Boers et al. found a median time to progression (TTP) of 16.8 months for patients with 100% FES-positive lesions, 6.2 months for patients with heterogeneous FES-positive lesions, and 3.5 months for patients without FES-positivity [24]. Liu et al. reported that mBC patients with only [18F]FES positive lesions had a mean PFS of 23.6 months, while patients with a [18F]FES negative site had a mean PFS of 2.4 months [63].

[18F]FES PET/CT can detect patients with ER heterogeneity who respond better to chemotherapy than endocrine therapy and thus prolong PFS. Xie et al. found that mBC patients with ER heterogeneity who received chemotherapy had a significantly higher median PFS compared to patients receiving endocrine therapy (median PFS 7.1 versus 4.6 months). Further to this point, the combination of chemotherapy and endocrine therapy did not improve PFS in this patient group (median PFS 4.4 months) [64]. These observations show that the optimal determination of the ER status can result in significant patient benefit.

The benefits of integrating [18F]FES PET/CT into the diagnostic workup of mBC and rBC patients, may have positive impacts beyond those already identified and associated with its current use (indication). If [18F]FES PET/CT was added to the standard diagnostic work up of mBC and rBC patients in line with the Appropriate Use Criteria (AUC) for ER targeted PET Imaging, the increase in diagnostic accuracy associated with [18F]FES PET/CT may improve the clinical outcomes for women associated with the 4 AUCs [17]. This is especially apparent when the current standard of care (biopsy and IHC) does not provide any ER assessment as is the case for women with mBC whose lesions are difficult to biopsy, when a biopsy is inconclusive, and when other imaging tests are equivocal or suggestive, i.e., clinical dilemmas. In addition, the increased accuracy with which clinicians could understand the functionality of ERs and their response to endocrine therapy, for both initially diagnosed mBC patients as well as after disease progression (rBC) may result in greater positive disease outcomes when a more appropriate treatment is undertaken than without [18F]FES PET/CT. Widespread adoption of [18F]FES PET/CT–or more broadly molecular imaging in evaluating disease characteristic may catalyze research and development and eventual use of increase diagnostic accuracy

associated with other molecular biomarkers like human epidermal growth factor receptor 2 (HER2) or progesterone receptor (PR) however, this is an area in need of additional study.

While there are limitations to increased use of molecular imaging in BC, these limitations are associated more with local regulations and realities linked to radiotracer preparation and transportation than access and affordability associated with different healthcare systems. Across different countries there are varying requirements that govern the production of molecular imaging agents. These rules dictate everything from the technology transfer to allow a third party to radiate a dose to the transport of radioactive materials thereby challenging their supply. Another limitation is the availability of PET machines on which to use the tracers. While this study has focused on the cost-effectiveness of [18F]FES PET/CT in the US, the outcomes should be transferable to other single Payer systems since the cost of the tracer has such a minimal impact on the (positive) outcomes (as evidence by the OWSA). This model-based economic analysis which is associated with an improvement in clinical and economic outcomes is subject to limitations. Among these limitations was the fact that the outcomes were not the result of a single dedicated study, but instead based on multiple sources. Due to this design, ER positivity is based on a threshold, i.e., >1% ER positivity, rather than individual ER values, and thus ER expression cannot be compared between true positive (IHC+/PET+) and false negative (IHC+/PET-) individuals. An additional consideration with the comparison between IHC and FES PET/CT is that IHC detects the presence of the ER on cells, while FES PET/CT detects the presence of functional receptors; thus, some "false negative" patients may have ER expression without ER functionality and unlikely to benefit from ER-directed treatment. Furthermore, this analysis assumed that the costs of the diagnostic procedures, tests and treatments will remain the same over time. Future analyses using real-world data may help address the uncertainties around model inputs.

## Conclusion

In conclusion, utilizing [18F]FES PET/CT in addition to biopsy/IHC for the determination of ER-positive status can be valuable for patients with mBC and rBC. The advantage of FES is particularly evident when a tumor sample cannot be obtained, or the risk of a biopsy-related complication is high. From the patients' and oncologists' perspective, the use of [18F]FES PET/CT in addition to biopsy/IHC may offer an increase in the diagnostic accuracy of the ER status, especially when multiple lesions are present. Benefits are expressed in the avoidance of repeated biopsies and futile treatments. Additionally, oncological centers can benefit from using [18F]FES PET/CT as it not only increases confidence in their actions but can also result in cost savings.

## Supporting information

**S1 Fig. Total costs per scenario for mBC patients when biopsy failed or was inconclusive.**
(DOCX)

**S2 Fig. Total costs per scenario for mBC patients when biopsy was not possible.**
(DOCX)

**S3 Fig. Total costs per scenario for rBC patients when [18F]FES PET/CT is used in combination with IHC.**
(DOCX)

**S4 Fig. Tornado Diagram: OWSA for the 10 most sensitive parameters in mBC patients when biopsy failed or was inconclusive.**
(DOCX)

**S5 Fig. Tornado Diagram: OWSA for the 10 most sensitive parameters in mBC patients when biopsy was not possible.**
(DOCX)

**S6 Fig. Tornado Diagram: OWSA for the 10 most sensitive parameters in rBC patients when [18F]FES PET/CT is used in combination with IHC.**
(DOCX)

**S7 Fig. Cost-effectiveness plane for mBC patients when biopsy failed or was inconclusive.**
(DOCX)

**S8 Fig. Cost-effectiveness plane for mBC patients when biopsy was not possible.**
(DOCX)

**S9 Fig. Cost-effectiveness plane for rBC patients when [$^{18}$F]FES PET/CT is used in combination with IHC.**
(DOCX)

**S1 Table. Results of the PSA (mean values).**
(DOCX)

## Acknowledgments

Editorial support was provided by Shereen Cynthia D'Cruz, Ph.D., of Certara Synchrogenix.

## Author Contributions

**Conceptualization:** Regina Munter-Young, Adolfo Fuentes-Alburo, Kurt Neeser.

**Data curation:** Regina Munter-Young, Kurt Neeser.

**Formal analysis:** Adolfo Fuentes-Alburo, Nicholas DiGregorio, Kurt Neeser.

**Funding acquisition:** Regina Munter-Young, Kurt Neeser.

**Investigation:** Adolfo Fuentes-Alburo, Kurt Neeser.

**Software:** Dmitry Gultyaev.

**Supervision:** Regina Munter-Young.

**Writing – original draft:** Regina Munter-Young, Nicholas DiGregorio, Kurt Neeser.

**Writing – review & editing:** Regina Munter-Young, Adolfo Fuentes-Alburo, Nicholas DiGregorio.

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
