## [Decision Letter · Decision Letter 0]

26 Jan 2024

PONE-D-23-43099Clinical and economic outcomes of adding 18F-FES PET/CT in estrogen receptor status identification in metastatic and recurrent breast cancer in the USPLOS ONE

Dear Dr. Munter-Young,

Thank you for submitting your manuscript to PLOS ONE. After careful consideration, we feel that it has merit but does not fully meet PLOS ONE’s publication criteria as it currently stands. Therefore, we invite you to submit a revised version of the manuscript that addresses the points raised during the review process.

We look forward to receiving your revised manuscript.

Kind regards,

Pierpaolo Alongi

Academic Editor

PLOS ONE

Journal Requirements:

"I have read the journal's policy and the authors of this manuscript have the following competing interests: RM-Y, AF-A, and ND are employees of GE HealthCare, which funded the health economic analysis. KN and DG are employees of Certara Germany GmbH, a consulting company that was paid by GE HealthCare for the services rendered."

We note that one or more of the authors are employed by a commercial company: GE HealthCare, Certara Germany GmbH

(2) Please also provide an updated Competing Interests Statement declaring this commercial affiliation along with any other relevant declarations relating to employment, consultancy, patents, products in development, or marketed products, etc.  

Within your Competing Interests Statement, please confirm that this commercial affiliation does not alter your adherence to all PLOS ONE policies on sharing data and materials by including the following statement: ""This does not alter our adherence to  PLOS ONE policies on sharing data and materials.” (as detailed online in our guide for authors http://journals.plos.org/plosone/s/competing-interests) . If this adherence statement is not accurate and  there are restrictions on sharing of data and/or materials, please state these. Please note that we cannot proceed with consideration of your article until this information has been declared.

Reviewers' comments:

Reviewer's Responses to Questions

**Comments to the Author**

1. Is the manuscript technically sound, and do the data support the conclusions?

Reviewer #1: Partly

Reviewer #2: Yes

2. Has the statistical analysis been performed appropriately and rigorously? 

Reviewer #1: Yes

Reviewer #2: Yes

3. Have the authors made all data underlying the findings in their manuscript fully available?

Reviewer #1: Yes

Reviewer #2: Yes

4. Is the manuscript presented in an intelligible fashion and written in standard English?

Reviewer #1: Yes

Reviewer #2: Yes

5. Review Comments to the Author

Reviewer #1: This manuscript endeavors to assess the financial implications and cost-effectiveness associated with the incorporation of the diagnostic technique [18F]FES PET/CT alongside biopsy/IHC in determining ER-positive status among patients with metastatic or recurrent breast cancer in the United States. The global significance of this paper lies in its clear demonstration of the synergistic role played by [18F]FES PET/CT and biopsy. It is desirable to adhere to the EANM nomenclature when referring to all radiotracers. In the Results section, tables require more detailed explanations within the text (e.g. Table 2 reports... etc.). Furthermore, the Discussion section should concentrate more on speculating about the role of [18F]FES PET/CT.

Reviewer #2: The manuscript is well-designed and provide important information regarding the cost saving of adding FES PET in the management of patients with breast cancer.

My comments are presented below:

- The methods needs an additional section or paragraph to describe how FES PET/CT has been carried out and how biopsy matched the PET/CT imaging.

- I suggest to add the mean expression of ER in true positive and false negative cases;

- I suggest to add the point that the procedure improves also the QOL (quality of life) of patients;

- The author may hypothise the economic impact in another economic scenario (e.g. Europe), where patients have no private medical insurance.

6. PLOS authors have the option to publish the peer review history of their article (what does this mean?). If published, this will include your full peer review and any attached files.

Reviewer #1: No

Reviewer #2: **Yes: **Natale Quartuccio

---

## [Author Response · Author response to Decision Letter 0]

8 Mar 2024

Reviewers' Comments to the Authors:

Reviewer 1

This manuscript endeavors to assess the financial implications and cost-effectiveness associated with the incorporation of the diagnostic technique [18F]FES PET/CT alongside biopsy/IHC in determining ER-positive status among patients with metastatic or recurrent breast cancer in the United States. The global significance of this paper lies in its clear demonstration of the synergistic role played by [18F]FES PET/CT and biopsy.

Author response: Thank you!

1. Comment from Reviewer 1 on the need to adhere to the EANM nomenclature when referring to all radiotracers. 

Author response: Thank you for highlighting this. The manuscript has been reviewed by all authors to confirm that the correct EANM nomenclature has been included. Reviewer 1 is correct, and we have adapted the naming of the radiotracers to the EANM nomenclature. More specifically, the first reference to FDG was changed to “2-deoxy-2-[18F]fluoro-D-glucose or 2-[18F]fluoro-2-deoxy-D-glucose” and each subsequent mention was cited as “[18F]FDG”. In addition, the first reference of FES was changed to “16α-[18F]fluoro-17β-fluoroestradiol” and “[18F]FES” for each successive reference. Finally “89Zr-trastuzumab” was changed to “[89Zr]Zr-Trastuzumab”

2. Comment from Reviewer 1 that tables in the Results section require more detailed explanations within the text.

Author response: As suggested, further explanations were added for all the tables and figures in the Results section. The revised text can be found on rows 209, 268 and 295.

[row 209] “Table 2 quantifies the economic, diagnostic, and clinical benefits of adding [18F]FES PET/CT to the diagnostic workup in mBC patients when a biopsy failed, was inconclusive, or not possible. Table 2 also quantifies the extent to which [18F]FES PET/CT in combination with IHC benefits rBC patients upon initial and over time.”

[row 268] “Introducing [18F]FES PET/CT to determine ER-positive status in mBC and rBC is cost effective as portrayed in Table 3.”

[row 295] “S2 shows the OWSA for the 10 most sensitive parameters in different subpopulations. Figure A identifies the parameters that impacts the ICER most among mBC patients when biopsy failed or was inconclusive. Similarly, Figure B illustrates the parameters that impact the ICER among mBC patients when biopsy was not possible, and Figure C shows them among rBC patients when [18F]FES PET/CT is used in combination with IHC.”

3. Comment from Reviewer 1 indicated that the Discussion section should speculate more about the role of [18F]FES PET/CT.

Author response: Thank you for the opportunity to expound on the benefits of FES. An entire section, beginning on row 386, has been added to the Discussion section highlighting additional benefits of integrating [18F]FES PET/CT into the diagnostic workup of mBC and rBC patients beyond those initially identified. In exploring these additional benefits, limitations associated with local radiotracer preparation, transportation and affordability were also included.

Reviewer 2

The manuscript is well-designed and provide important information regarding the cost saving of adding FES PET in the management of patients with breast cancer.

Author response: Thank you for the positive feedback.

1. Comment from Reviewer 2 to add a section or paragraph to the Methods section to describe how FES PET/CT has been carried out.

Author response: This important element was, in fact, omitted from the initially submitted manuscript and has been corrected in the updated version. The production of FES and a description of the administration of FES and PET/CT scan are outlined beginning on line 144 of the updated manuscript.

[row 144] “[18F]FES was produced locally as described by Roemer J, et al. and Venema CM et al. [49, 50] The [18F]FES PET/CT scans were performed on a BioGraph mCT 40- or 64-slice PET/CT (Siemens/CTI, Knoxville, TN) or an Ingenuity TF or Gemini TF PET/CT (Philips Medical Systems, Cleveland, OH) where about 200 MBq [18F]FES were injected intravenously as a bolus. As described by van Geel et al., PET/CT images were acquired 60 ± 10 minutes after tracer injection. Before the PET scan, a low-dose CT was conducted for attenuation and scatter correction. A whole-body PET emission scan was obtained with a 3-minute per bed position acquisition time. All patients were scanned from head to mid-thigh and for all scan reconstructions and quantiﬁcations, the European Association of Nuclear Medicine Research Limited (EARL) criteria were followed. A full account of [18F]FES PET/CT whole body evaluation and the quantity of FES taken up by the metastases is further described in by van Geel et al.”

2. Comment from Reviewer 2 to update the Methods section to describe how biopsy matched the PET/CT imaging.

Author response: An addition to the Methods section beginning on row 167 of the update manuscript explains the meta-analyses used to compare the diagnostic accuracy of biopsy/IHC and [18F]FES PET/CT

3. Comment from Reviewer 2 to add the mean expression of ER in true positive and false negative cases.

Author response: This was an oversight in the original manuscript, for which, again we thank the reviewer for this important feedback. A description of how the biopsy/IHC ER status matched those of the [18F]FES PET/CT were updated in Table 1, where the rates of true ER positive and ER negative were reported when an ER expression threshold of >1% was applied.

4. Comment from Reviewer 2 suggesting an addition about how the procedure improves patients’ QOL.

Author response: The quantifiable increase in patients’ QoL is very small and as such only a small change was made to the manuscript to reflect this on row 350.

“This study therefore suggests that adding [18F]FES PET/CT to biopsy/IHC could have a positive impact on clinical and economic outcomes as well as on patients´ quality of life (QoL) in the US.”

5. Comment from Reviewer 2 to hypnotize on the economic impact that FES may have in other economic scenarios (e.g., Europe), where patients have no private medical insurance.

Author response: Differences in local regulations that govern the production of radioactive diagnostics (outlined beginning on row 403), the cost to manufacture and distribute them may impact their acquisition price and subsequently the actual economic benefits; however, based on the results of the OWSA since the cost of the tracer has such a minimal impact on the cost-effectiveness, the cost-effectiveness of [18F]FES PET/CT as calculated in the US can be readily transferred to other payer archetypes.

“While there are limitations to increased use of molecular imaging in BC, these limitations are associated more with local regulations and realities linked to radiotracer preparation and transportation than access and affordability associated with different healthcare systems. Across different countries there are varying requirements that govern the production of molecular imaging agents. These rules dictate everything from the technology transfer to allow a third party to radiate a dose to the transport of radioactive materials thereby challenging their supply. Another limitation is the availability of PET machines on which to use the tracers. While this study has focused on the cost-effectiveness of [18F]FES PET/CT in the US, the outcomes should be transferable to other single Payer systems since the cost of the tracer has such a minimal impact on the (positive) outcomes (as evidence by the OWSA).”

---

## [Editor Report · Decision Letter 1]

5 Apr 2024

Clinical and economic outcomes of adding [18F]FES PET/CT in estrogen receptor status identification in metastatic and recurrent breast cancer in the US

PONE-D-23-43099R1

Dear Dr. Munter-Young,

We’re pleased to inform you that your manuscript has been judged scientifically suitable for publication and will be formally accepted for publication once it meets all outstanding technical requirements.

Kind regards,

Pierpaolo Alongi

Academic Editor

PLOS ONE
---

## [Editor Report · Acceptance letter]

26 Apr 2024

PONE-D-23-43099R1 

PLOS ONE

Dear Dr. Munter-Young, 

I'm pleased to inform you that your manuscript has been deemed suitable for publication in PLOS ONE. Congratulations! Your manuscript is now being handed over to our production team.

Kind regards, 

on behalf of

Dr. Pierpaolo Alongi 

Academic Editor

PLOS ONE